# Improving Printability of Polytetrafluoroethylene (PTFE) with the Help of Plasma Pre-Treatment

**DOI:** 10.3390/polym15163348

**Published:** 2023-08-09

**Authors:** Marius Andrei Olariu, Rakel Herrero, Dragoș George Astanei, Lisandro Jofré, Javier Morentin, Tudor Alexandru Filip, Radu Burlica

**Affiliations:** 1Faculty of Electrical Engineering, “Gheorghe Asachi” Technical University of Iasi, 700050 Iasi, Romania; molariu@tuiasi.ro (M.A.O.); tudor-alexandru.filip@academic.tuiasi.ro (T.A.F.); 2Technological Center of Mobility and Mechatronics (NAITEC), 31200 Estella-Lizarra, Spain; rherrero@naitec.es (R.H.); ljofre@naitec.es (L.J.); jmorentin@naitec.es (J.M.)

**Keywords:** polytetrafluoroethylene (PTFE), plasma pre-treatment, functional printing

## Abstract

Polytetrafluoroethylene (PTFE) is a potential candidate for the fabrication of flexible electronics devices and electronics with applications in various extreme environments, mainly due to its outstanding chemical and physical properties. However, to date, the utilization of PTFE in printing trials has been limited due to the material’s low surface tension and wettability, which do not ensure good adhesion of the printing ink at the level of the substrate. Within this paper, successful printing of PTFE is realized after pre-treating the surface of the substrate with the help of dielectric barrier discharge non-thermal plasma. The efficiency of the pre-treatment is demonstrated with respect to both silver- and carbon-based inks that are commercially available, and finally, the long-lasting pre-treatment effect is demonstrated for periods of time spanning from minutes to days. The experimental results are practically paving the way toward large-scale utilization of PTFE as substrate in fabricating printed electronics in harsh working environments. After 3 s of plasma treatment of the foil, the WCA decreased from approximately 103° to approximately 70°. The resolution of the printed lines of carbon ink was not time dependent and was unmodified, even if the printing was realized within 1 min from the time of applying the pre-treatment or 10 days later. The evaluation of the surface tension (σ) measured with Arcotest Ink Pink showed an increase in σ up to 40 < σ < 42 mN/m for treated Teflon foil and from σ < 30 mN/m corresponding to the untreated substrate. The difference in resolution was distinguishable when increasing the width of the printed lines from 500 μm to 750 μm, but when increasing the width from 750 μm to 1000 μm, the difference was minimal.

## 1. Introduction

Organic flexible electronics emerged as high potential devices that are to reshape the world of electronics due to their easy integration potential in any sector, from health [1] to industry [2], agriculture [3], energy harvesting [4], automotive [5], and even the space industry [6]. These types of devices are developed at the level of bendable substrates (e.g., polymers and/or paper) on top of which one or multiple layers of functional printed materials are patterned via printing techniques (e.g., screen printing, ink-jet, flexography, gravure) [7]. Despite being cost-efficient from a manufacturing viewpoint, flexible electronics have the disadvantage of being developed on very economical polymeric substrates (e.g., polyesters) [8], materials that are susceptible to rapid degradation when operating in extreme or harsh working conditions (e.g., high temperature) [9]. To address this disadvantage, fluoropolymers were considered potential candidates for developing flexible electronics for operating within severe working environments. Fluoropolymers are a category of high-performance technical polymers with superior mechanical, chemical, and thermal properties. These encompass materials such as polytetrafluoroethylene (PTFE or commercially known as TeflonTM), fluorothermoplastics, and fluoroelastomers because of their strong C-F bonds and weak polarity [10]. However, despite their outstanding properties, fluoropolymers do not exhibit convenient properties for being directly printed because of their wettability, low surface tension, and contact angle, which are crucial parameters in ensuring the optimal transfer and solidification of liquid inks or pastes at the surface of the printing substrate. The issue is common in the case of flat printing techniques, but also in the case of roll-to-roll ones, and should be addressed. 

Of all fluoropolymers, polytetrafluoroethylene (PTFE) is perhaps the most well known due to its excellent chemical resistance and very good behavior in extreme working environments. Commonly, PTFE is used as an electrical insulator in applications requiring stable properties at high temperatures due to its very good dielectric properties [11]. Electrical circuits printed on PTFE can have a multitude of applications, especially given the thermal resistance of the foil and the very good dielectric rigidity of the polymer material. Applications can be imagined, beginning with temperature microsensors and thermochemical or printed thermoresistors. However, when PTFE is considered as a potential candidate for flexible electronics, common printing problems encountered in the case of fluoropolymers should not be neglected because of the polymer’s low surface tension, wettability, and large water contact angle (WCA). In order to address these technological problems, a series of solutions has been proposed within the literature. Chemical treatment techniques were first employed to improve the surface properties of PTFE and make it suitable for printing applications [12].

Wettability, which is influenced according to Golub [13] by the distribution of acidic groups and/or surface dipoles across the interface between the PTFE surface and liquid (ink), is a property of high importance with respect to the solid–liquid interaction due to its influence in practical applications in coating, flotation, mass transport, catalysis, chemical reactions at the solid–liquid interface, agrochemicals, flows in reservoirs, nanofabrication, batteries, and separation [14]. The chemical solutions proposed previously for improving wettability demonstrated a series of drawbacks, such as potential high toxicity and being hazardous to the environment, as well as the appearance modifications it induced [15]. To address these disadvantages, Lojen employed low-pressure non-equilibrium inductively coupled RF hydrogen plasma in the H-mode in order to functionalize the surface of PTFE with polar groups in order to improve the adherence of various inks. On the other hand, in [16], the increased adhesion of the PTFE surface improved hardness, and was obtained with the help of heat-assisted plasma treatment at atmospheric pressure. 

Atmospheric pressure non-thermal plasma represents a sustainable and eco-friendly alternative technology for surface treatment compared to other methods, such as chemical treatment, electrochemical processes, or flame treatment [17]. Non-thermal plasma can be used for polymer surfaces in order to improve their properties, such as adhesion, surface tension, wettability, and printability [18,19]. Non-thermal plasma treatments have been applied to various types of polymeric substrates, such as polycarbonate, fluorinated ethylene propylene, polyimide, polyethylene naphthalate, polyethylene terephthalate, [20], polypropylene [21], polycarbonate [22], and polytetrafluoroethylene [23]. The results reported in the above-mentioned paper indicate that plasma treatment can ensure an improvement in polymeric surface wettability and increased surface tension. AFM analysis indicated an increase in the adhesion force and surface roughness, while XPS analyses indicated an increase in oxygen content in polymeric film surfaces. The most common non-thermal plasmas used for polymeric film surfaces are corona and dielectric barrier discharge (DBD). The technologies that use non-thermal plasma are simple and easy to apply in practice due to the fact that they work in ambient conditions, i.e., temperature and pressure, so the investments in their implementation in practice are minimal. Non-thermal plasma represents an ionized gas medium in thermodynamic imbalance. In this ionized medium, the free electrons produced by an electric discharge have a very high temperature > 5000 K, while the plasma gas is found at room temperature. The non-thermal character of the plasma can be determined using different conditions of the evolution of the electric discharge, either by interposing a dielectric substrate (quartz) between the electrodes, the electrical discharge with the dielectric barrier (DBD), or by forcing the evolution of the electric discharge in a gas flow (air), in which case the speed of the gas maintains the thermodynamic imbalance of the electric discharge. Although corona plasma (an electrical discharge of a low power point to plan at the wire to plan the configuration) is widely used, and its effectiveness in surface modification has been proven, it presents the disadvantage of generating inhomogeneous plasma on relatively reduced areas [24]. To avoid the corona treatment drawbacks, DBD plasma represents an effective alternative [25]. 

In line with the above-mentioned problems and manufacturing needs, this paper presents a series of laboratory experiments demonstrating the effectiveness of DBD non-thermal plasma pre-treatments for ensuring the improved long-term wettability of PTFE used in both the screen and ink-jet printing processes of printed or flexible electronics. This paper focuses on the results obtained when treating the PTFE substrate surface using a planar DBD reactor operating under atmospheric conditions (pressure, temperature, humidity). 

## 2. Materials and Methods

PTFE substrates, commercially available under the name of DuPont™ Teflon^®^ FEP (DuPont, Wilmington, DE, USA), were employed in the experiments. The substrates’ thickness was 76 μm, while the dielectric strength was 260 kV/mm for the 0.025 mm film, which, is mentioned within the technical fiche of the material provided by the manufacturer. 

The plasma pre-treatments were performed with the help of an in-house manufactured non-thermal plasma generator that allows for the generation of an output AC voltage in the range of 7 kV with a frequency of approximately 21.5 kHz, with the average power per discharge being 40 W for a 6 mA current. The construction of the non-thermal plasma generator is presented within our previous work [26].

In the experimental setup presented in Figure 1, the DBD plasma reactor consists of two electrodes (E1 and E2) connected to a power supply HVPS 0–20 kV/20–70 kHz (PVM500-2500-Plasma Power Generator, Denver, CO, USA). Between the electrodes, two 3 mm-thick glass sheets are placed. The distance between the electrodes is approximately 9 mm.

The high voltage is measured, as indicated in Figure 1, through an HV voltage probe 1000:1, and the current value is taken on a 100 Ω shunt using a digital LeCroy WaveSurfer 3000 oscilloscope (Teledyne, www.teledynelecroy.com, 13 July 2023). The voltage and current waveforms are presented in Figure 2.

The recorded voltage and current waveforms were used to determine the power of the electrical discharge.

PTFE substrates were then treated with the help of the DBD plasma. The Teflon substrates involved in the experiments were treated with DBD plasma for 5 s. The printing trials with silver ink were run immediately after the substrates were treated (within the first minute) and were repeated on treated substrates after 60, 120, 180, and 240 min in order to evaluate the plasma effects persistence over time. Imagistic observations were performed at the level of straight-printed lines of different widths (200 μm, 500 μm, 750 μm, and 1000 μm, respectively). A similar working procedure was established for printing treated substrates with carbon ink. The screen-printing experiments with carbon ink were performed within 1 min after applying the treatment and repeated 10 days after submitting the substrate to plasma treatment. In order to avoid ambient contamination during the 10-day depositing period, the treated PTFE substrates were introduced into vacuum-sealed polymeric bags. As in the case of the silver screen-printing experiments, the printing trials with carbon were performed at the level of straight-printed lines of different widths (200 μm, 500 μm, 750 μm, and 1000 μm, respectively). 

For the printing experiments, three techniques were employed: flat-screen printing, ink-jet printing, and R2R (roll-to-roll) screen printing. 

Screen printing and ink-jet printing, from a technological viewpoint, are two non-contact printing methods that assume the transfer of a pattern at the level of a substrate. 

The first of these, screen printing, is perhaps the oldest and simplest technique, which was adapted from printing to performing the functional printing of electrical and electronic devices. The technology facilitates the transfer of a certain material, in a liquid state, at the level of the substrate with the help of a screen mesh attached to the screen mask containing openings positioned under the shape of the image to be transferred. The printed pattern is realized by solidification of the so-called ink or paste, which is transferred at the surface of the substrate through the openings of the screen mesh.

On the other hand, the ink-jet printing method is an automatized process supposing the creation of a desired pattern on the surface of a substrate through the controlled disposal of ink microdroplets in accordance with the digital image/shape elaborated by the human operator. As is known, this printing method was first used in the publishing industry and was afterwards adopted by office printing devices and the flexible electronics industry.

In spite of the fact that in both cases the material to be transferred should be liquid, the ink employed within ink-jet printing should exhibit much lower viscosity, but in both cases, the ink’s adherence to the substrate is a fundamental property. 

Within our work, the flat screen-printing experiments were performed with a semiautomatic Ekra E2XL printer (Figure 3–left) by employing a 325 stainless steel screen mask with a 25 μm EOM (emulsion on mask) and a diamond-shaped squeegee. The screen-printing process variables were the following: off-contact distance 2 mm, pressure 1.5 N, squeegee speed 80 mm/s, curing temperature 120 °C, and curing speed 0.95/min. 

The ink-jet printing single layer was realized using an Ardeje Origin D100 ink-jet printer (Figure 3–right) at 600 dpi resolution and a speed of 100 mm/s in the multi-pass printing mode, while curing was realized for 1 h at 140 °C. 

The R2R experiments were performed with the help of Lambda, Edale R2R machinery, (Figure 4) for which the process parameters were the following: printing speed of 5 m/min and curing temperature of 120 °C. For cost-effective reasons, in the case of R2R printing, the paper roll was used as a carrier substrate, while the Teflon substrates were attached on top of it.

The water contact angle (WCA) and surface free energy were assessed with the help of an Ossila Contact Angle Goniometer in accordance with the ASTM D5946 standard [27], while surface tensions, σ, were determined with the help of Arcotest Ink Pink from 30 to 60 mN/m and accuracy +/−0.5 mN/m. The WCA and surface tension measurements were performed in the first minute after plasma treatment.

The printed layers were imaged with the help of a Nikon AZ100 microscope (Tokyo, Japan), while the profilometric analysis was performed with a BRUKER DEKART XT profilometer. 

Screen-printing carbon and silver inks were employed to perform the printing experiments. Both silver (SCAG-004) and carbon inks (SRC-012) were provided by Mateprincs. The technical characteristics of both inks are available on the producer’s commercial website. (www.mateprincs.com (accessed on January 2023)) For the ink-jet printing sessions, PV Nanocell (Migdal Ha’Emek, Israel) and Novacentrix JS-A211 (Austin, TX, USA) inks were used.

## 3. Results and Discussion

The screen-printing experiments with both silver and carbon inks were performed initially at the level of untreated Teflon substrates. The printing experiments realized at the level of untreated PTFE highlighted, as expected, the impossibility of printing this type of material with both inks because of PTFE’s wettability and surface tension, which do not allow ink to adhere to the material’s surface (Figure 5). 

As can be noticed, drops of inks formed at the substrate’s surface immediately after printing due to polar groups, which are present at the interface between the polymer and the printing ink.

The evaluation of the surface tension (σ) measured with Arcotest Ink Pink showed an increase in σ up to 40 < σ < 42 mN/m for the treated Teflon foil and from σ < 30 mN/m corresponding to untreated substrate. The WCA evaluated after plasma treatment for the treatment indicated a decrease from approximately 103° to approximately 70° after treatment times of 3 s, as reported in Figure 6. The increase in treatment time had a small influence on WCA, with the values obtained for 30 s of plasma treatment being approximately 65°.

### Evolution of Ink Adhesion on Treated/Untreated Film

The improvement in printing conditions on PTFE are indubitably visible. The plasma pre-treatment allowed the patterning of a continuous layer of silver ink on the entire pre-treated surface. One can attribute the successful printing of silver ink to a considerable improvement in both wettability and surface tension during the pre-treatment processes because of the peroxide radicals and functional groups formed, which also considerably enhance the adhesion of silver layers (see Figure 7).

As can be seen in Figure 7, in the area not treated with non-thermal plasma, the conductive ink has a tendency to regroup in a multitude of microdroplets, with the printed circuit obtained being practically unusable. Instead, on the treated surface, the earring has a uniform appearance, which indicates quality printing of the electrical circuit. Moreover, it is noticeable that the effect of plasma pre-treatment lasted for longer periods of time, a fact which is emphasized by the outstanding printing results obtained when running the experiments after 240 min from the moment when the treatment was performed, as can be seen in Figure 8.

As can be noticed in Figure 8, the resolution of the printed lines of silver ink improves while the width of the line increases. The resolution improvement is evident when comparing the quality of the line’s edges of 200 μm width with the resolution of those of 500 μm width. While increasing the width to 750 μm and 1000 μm, the differences in resolution are minimal and almost invisible. This fact denotes once again the fact that the plasma pre-treatment has a long-lasting period.

The situation is similar with respect to the screen-printing experiments with the carbon-based ink at PTFE’s surface. As can be noticed, a layer of continuous carbon ink was successfully patterned at the level of the PTFE substrate surface immediately after the pre-treatment was applied. The screen-printing trials performed 10 days after pre-treating the sample demonstrate that the surface physical and chemical modifications are stable over time, in which case contact with the ambient environment is avoided.

The resolution of the printed lines of carbon ink is not time-dependent and is unmodified, even if the printing is realized within 1 min from the time of applying the pre-treatment or 10 days later. The difference in resolution is distinguishable while increasing the width of the printed lines from 500 μm to 750 μm, but when increasing the width from 750 μm to 1000 μm, the difference is minimal (see Figure 9).

In terms of the width of the patterned traces of silver ink, the average values obtained are presented below. The average values of the width measured at the level of the silver screen-printed lines are very close to the expected width dimensions (Table 1).

Resistivity measurements were also performed after the treatment of the substrates. Samples were treated for 5 s and printed using flat-screen printing at 0, 1, 2, 3, and 4 h after treatment. As can be seen, there were no significant differences in respect to resistivity values over time; thus, one can state that printing the samples up to 4 h after pre-treatment is possible from the point of view of conductivity values (Table 2).

According to profilometric determinations and analysis, it can be noticed that the thickness of the ink patterned at the level of the substrate is not dependent on the time when the printing was performed. 

However, a small difference was noticed in respect to the thickness of the layer. The expected height of the layer, according to our design and emulsion thickness on the mesh, is approximately 15 μm, while the one measured at the level of the layer patterned immediately after applying the pre-treatment is 16.60 μm. The height of the layer printed after 4 h was approximately 15.50 μm. However, a difference was noticed with respect to the height of the layer at the middle of the width. For the layer printed after 4 h, the height in the middle of the width was lower. Moreover, the height of the carbon layer measured is 8.82 μm, which is approximately 12% of the expected height according to our design and the thickness of the emulsion on the mesh (see Figure 10 and Figure 11).

The efficiency of the plasma pre-treatment was also evident in the case of the ink-jet printing experiments. As visible in Figure 12, ink-jet printing is impossible to realize at the level of untreated PTFE. After pre-treating the substrate with non-thermal plasma under similar conditions as those employed for the substrates used in the screen-printing experiments, printing is possible, and ink traces of different widths (200 μm, 500 μm, 750 μm, 1 mm, 1.5 mm, 2 mm, 2.5 mm, and 3 mm) are successfully printed. In Figure 12a,b, the effect of plasma treatment is visible on the right side of the picture. On the left side of the substrate, in the non-treated area, it is evident that the printing could not be realized, and the inks are not adherent to the substrate’s surface.

After curing the substrate at 140 °C, the resistance values of the printed traces were determined, and the values obtained are presented below (Table 3):

In order to verify the efficiency of the plasma treatment over time, ink-jet printing trials were performed after 120 min, 180 min, and 7 days, respectively. In the case of the experiments performed after 7 days, the substrates were deposited exactly in the same manner as in the case of the substrates employed in the screen-printing experiments. Our ink-jet printing experiments demonstrate the fact that the treatment lasts for long periods of time (Figure 13).

Last but not least, we verified the effect of plasma treatment by performing multilayered ink-jet printing. Thus, at the level of the pre-treated Teflon substrate, we patterned three layers of silver-based ink via ink-jet printing. Printing was also possible in the case of multilayered ink-jet printing trials, and our results are presented in Figure 14.

Overall, the printing trials (flat screen and ink-jet) and results obtained at the level of plasma-treated PTFE substrates clearly demonstrate the efficiency of plasma treatment with respect to improving the adhesion of silver- and carbon-based inks, but they more importantly demonstrate the fact that the effect of plasma treatment at the level of PTFE is less dependent on time. The increase in the surface tensions of PTFE from less than 30 mN/m to values above 40 mN/m seems to be sufficient for allowing the patterning of conductive inks at its surface. The improvement in surface tensions may be attributed to peroxide radicals (C–O–O) and active functional groups containing oxygen (O–C=O, C=O, C–O) as stated by [16].

## 4. Conclusions

PTFE is a polymer demonstrating outstanding chemical, mechanical, and thermal properties and might be a potential candidate for the development of flexible electronics for operating in extreme working environments, but its usability is hampered by difficult and low printing properties. The effect of non-thermal plasma pre-treatment at the level of PTFE substrates was evaluated from the printability capacity point of view. The printing trials (ink-jet and screen printing) were performed immediately after plasma pre-treatment and over longer periods of time (which spanned from minutes to days), with the aim of evaluating the time persistence of the pre-treatment. Significantly enhanced adherence of both the silver and carbon inks in the case of both types of printing technologies, namely ink-jet printing and screen printing, was encountered. The improvement in adhesion properties may be attributed to modifications in surface properties, which are mainly attributed to the formation of polar groups at the solid–liquid interface.

## Figures and Tables

**Figure 1 polymers-15-03348-f001:**
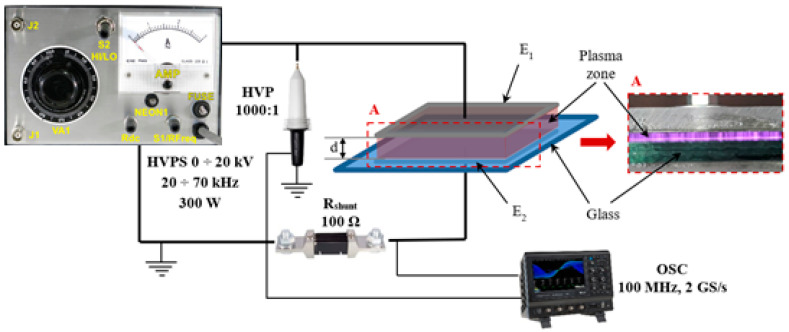
Experimental setup.

**Figure 2 polymers-15-03348-f002:**
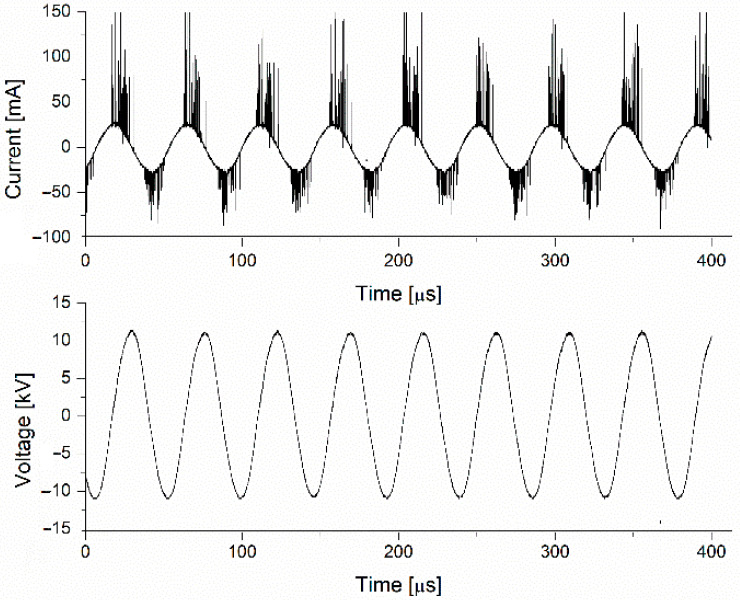
The voltage and current waveforms.

**Figure 3 polymers-15-03348-f003:**
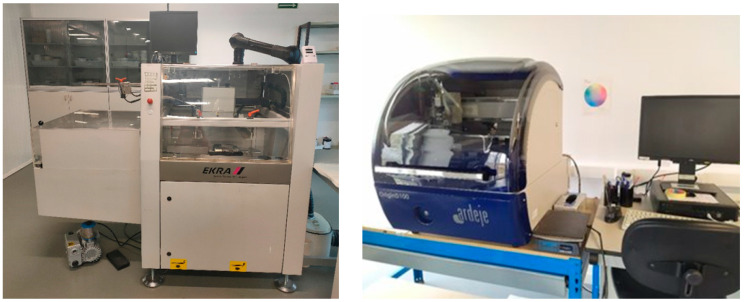
**Left**: Flat screen printer; **right**: ink-jet printer.

**Figure 4 polymers-15-03348-f004:**
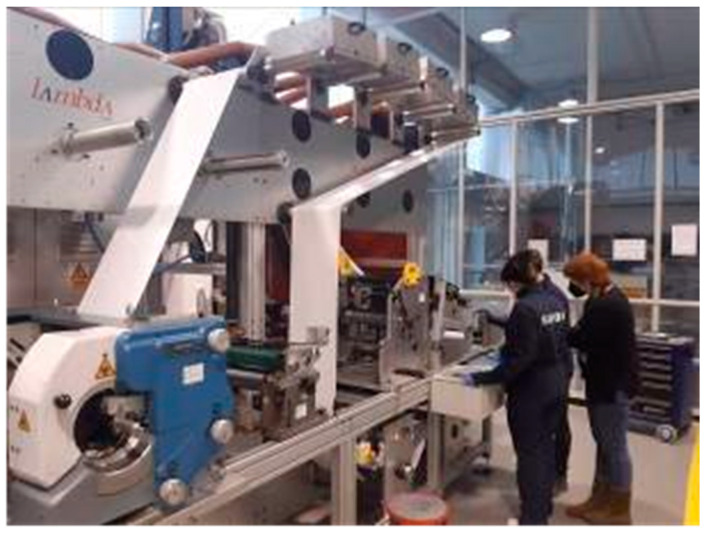
Preparation of rotary screen printer for tests.

**Figure 5 polymers-15-03348-f005:**
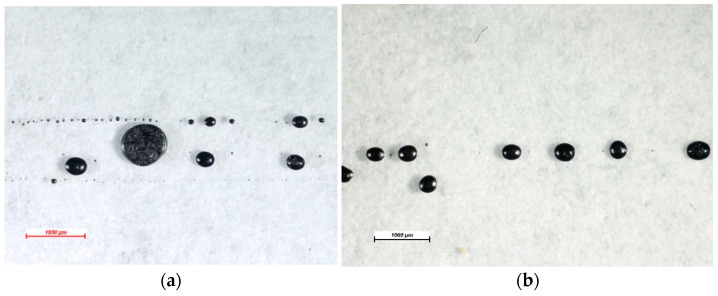
Untreated Teflon substrate screen printed with (**a**) silver ink and (**b**) carbon ink.

**Figure 6 polymers-15-03348-f006:**
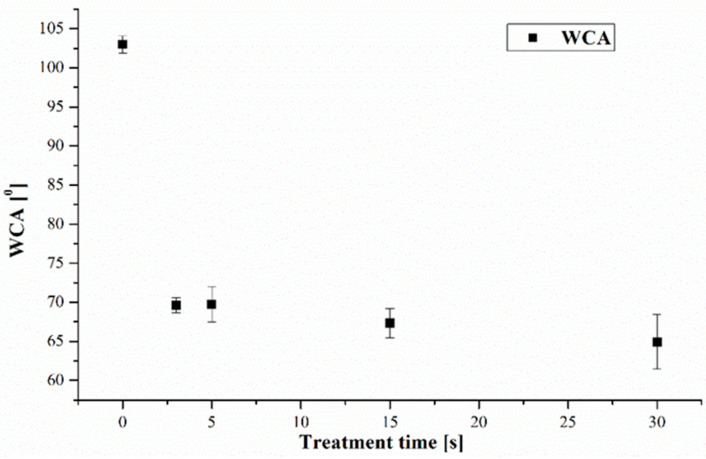
Evolution of WCA with plasma treatment time.

**Figure 7 polymers-15-03348-f007:**
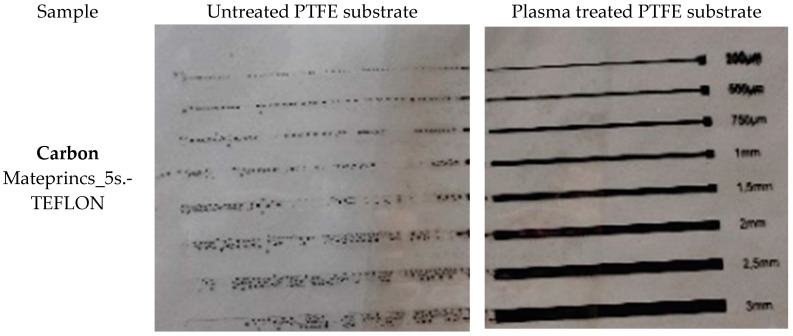
Screen-printed treated Teflon substrate with (**a**) silver ink and (**b**) carbon ink.

**Figure 8 polymers-15-03348-f008:**
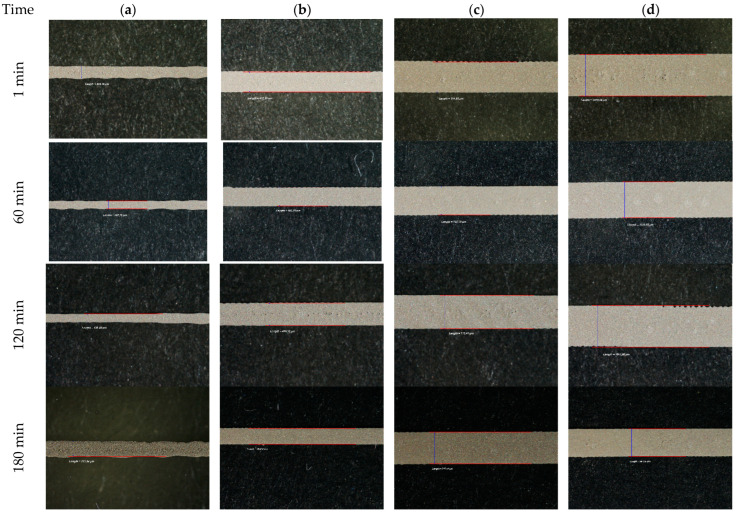
Effects of plasma pre-treatment on silver screen-printing experiments on Teflon after various times: (**a**) lines of 200 μm width, (**b**) lines of 500 μm width, (**c**) lines of 750 μm width, and (**d**) lines of 1000 μm width.

**Figure 9 polymers-15-03348-f009:**
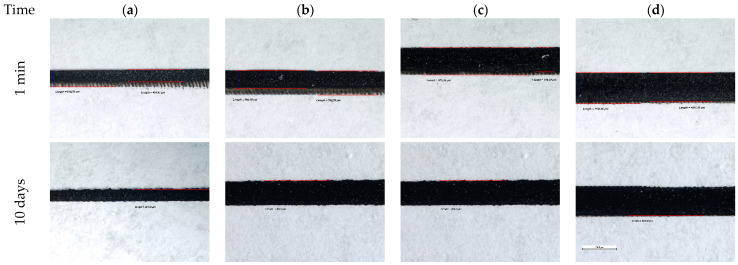
Effects of plasma pre-treatment on carbon screen-printing experiments on Teflon after various times: (**a**) lines of 200 μm width, (**b**) lines of 500 μm width, (**c**) lines of 750 μm width, and (**d**) lines of 1000 μm width.

**Figure 10 polymers-15-03348-f010:**
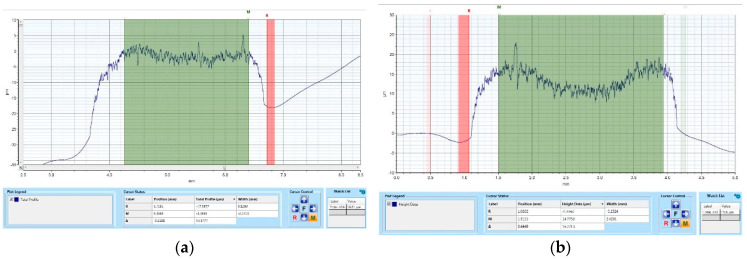
Effects of plasma pre-treatment on silver layer’s height screen printed on Teflon after various times: (**a**) 1 min after applying the pre-treatment, (**b**) 240 min after applying the plasma pre-treatment.

**Figure 11 polymers-15-03348-f011:**
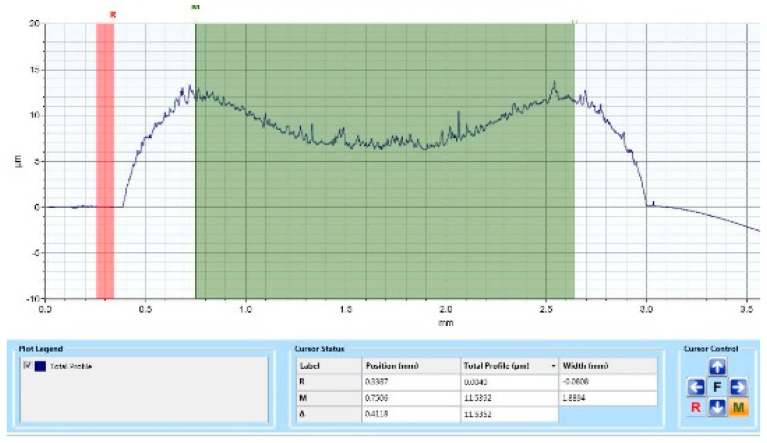
Effects of plasma pre-treatment on carbon layer’s height screen printed on Teflon, 1 min after applying the pre-treatment.

**Figure 12 polymers-15-03348-f012:**
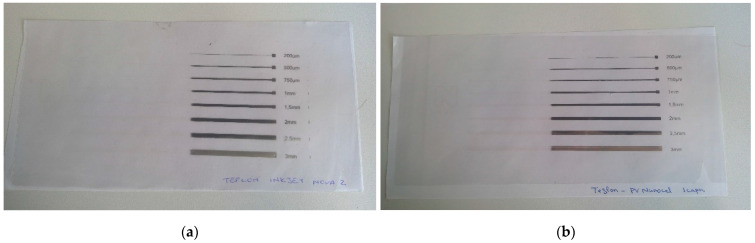
Ink-jet printed silver lines on Teflon substrate with (**a**) silver ink 1 and (**b**) silver ink 2.

**Figure 13 polymers-15-03348-f013:**
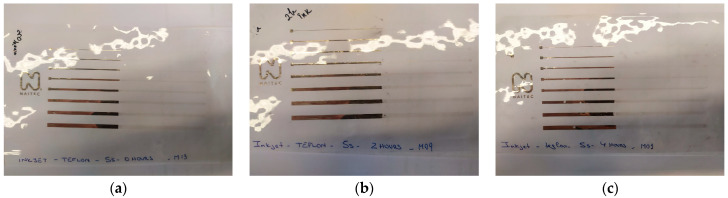
Ink-jet printed silver lines on Teflon substrate patterned after (**a**) 120 min, (**b**) 180 min, and (**c**) 7 days. The right side of each foil was plasma-treated, while the right side was untreated.

**Figure 14 polymers-15-03348-f014:**
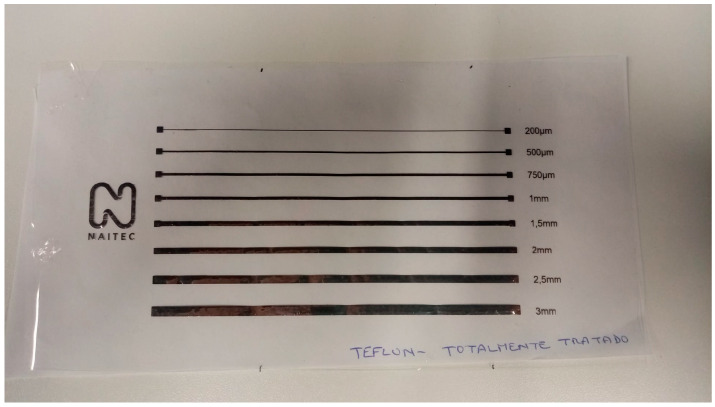
Multilayered ink-jet-printed silver lines on Teflon plasma-treated substrate.

**Table 1 polymers-15-03348-t001:** Average measured width of the silver screen-printed layers.

	200 μm	500 μm	750 μm	1000 μm
240 min	233.65	536.63	764.69	959.61
180 min	213.94	494.58	747.11	994.65
120 min	185.49	485.12	713.41	1011
60 min	207.15	486.18	727.15	1022.55
1 min	205.46	497.35	741.95	1015.08
Average value	209.14	499.97	738.86	1000.58

**Table 2 polymers-15-03348-t002:** Resistance values of silver traces flat-screen printed after treatment.

	Resistance (Ω)							
	200 μm	500 μm	750 μm	1000 μm	1500 μm	2000 μm	2500 μm	3000 μm
Ag Mateprincs_5s, 4 h	55.3	18.3	13	10.3	7.6	6.3	5.4	4.7
Ag Mateprincs_5s, 3 h	63	21.7	14.1	11.2	8.2	6.7	5.7	4.9
Ag Mateprincs_5s, 2 h	62	21.4	15	11.4	8.1	6.7	5.7	4.9
Ag Mateprincs_5s, 1 h	63.9	20.6	13.9	10.8	7.7	6.4	5.3	4.5
Ag Mateprincs_5s, 0 h	55.5	20.4	14.1	11.2	8	6.6	5.7	4.9

**Table 3 polymers-15-03348-t003:** Resistance values of silver traces ink-jet printed after treatment.

Trace Width (μm)	500 μm	750 μm	1000 μm	1500 μm	2000 μm	2500 μm	3000 μm
Resistance (Ω)	1127	643	375	214	216	203	145

## Data Availability

The data presented in this study are available on request from the corresponding author. The data are not publicly available due to partners’ IPR policy.

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
