# Peer review of "Improving Printability of Polytetrafluoroethylene (PTFE) with the Help of Plasma Pre-Treatment"

_polymers, 2023, doi:10.3390/polym15163348_

Round 1
Reviewer 1 Report
The surface properties changing of PFTE in barrier discharge are describes in the paper "Improving printability of Polytetrafluoroethylene (PTFE) with the help of plasma pre-treatment". There are results of printing the electrical conductors on the surface of PTFE, modification in barrier discharge are considered in that paper. The paper undoubtedly has practice interest. But there are some questions, some of them listed below:
1. The Introduction part can be expanded. Thus, it is known that the PFTE substrate is widely used in microwave technology.
2. The authors should clarify the measurement errors of the PFTE surface profile on the profilometer.
Author Response
Dear Reviewer,
Thank you for your time allocated for reviewing our manuscript! Please find below our responses to your comments:
- The Introduction part can be expanded. Thus, it is known that the PFTE substrate is widely used in microwave technology.
We expanded a bit the introduction of the manuscript. The abstract has been improved with a series of quantitative information, while the main part of the Introduction with some info on various applications and explanations regarding the non-thermal plasma treatments in general.
- The authors should clarify the measurement errors of the PFTE surface profile on the profilometer.
Unfortunately, we do not understand to what errors your comment is referring to. The profile of the printed traces are normal, higher on the edges and smaller at the inside as of Coffee ring or Maragoni effect. These are common effects noticeable at the level of printed traces and in our opinion the dimensional differences are more than reasonable. Hope that our reply is in line with your question! The spikes on the profiles may be attributed to printing mesh resolution, printed surface porosity and/or curing process!
Hope that our comments fulfills your expectations!
Regards,
Marius

Reviewer 2 Report
1- Include some important quantitative results in the abstract.
2- Any standard or reference for WCA measurement?
3- Add information such as trade name, manufacturer, city and country for all materials used in the study such as carbon and silver inks etc.
4- How did you measure the adhesion between the inks and the substrate? The method and obtained results should be explained in detail.
5- SEM, FTIR, AFM etc. are required to show the physical and chemical changes of the surface and explain the obtained improvements in printing results.
Minor revisions recommended.
Author Response
Dear Reviewer,
Thank you for your time allocated for reviewing our paper! Please find below our responses:
- Include some important quantitative results in the abstract.
Additional quantitative info has been included within the abstract as requested by the reviewer!
- Any standard or reference for WCA measurement?
Reference to the most know standard, ASTM D5946: Contact Angle Measurements ASTM D5946 / Test Method For Corona-Treated Polymer Films Using Water Contact Angle Measurements ASTM D5946 has been included within the paper
- Add information such as trade name, manufacturer, city and country for all materials used in the study such as carbon and silver inks etc.
Information regarding the inks are included within the manuscript text, as follows
„Both silver (SCAG-004) and carbon inks (SRC-012) were provided by Mateprincs”…
„For the ink-jet printing sessions, PV Nanocell and Novacentrix JS-A211 inks were used” banuiesc ca tot de Mateprincs sunt produse. Eu nu cred ca este cazul sa mai completam aici, se subintelege.
“PTFE substrates, commercially available under the name of DuPont™ Teflon® FEP, were employed in the experiments. The substrates’ thickness was of 76 μm while the dielectric strength of 260 kV/mm for 0.025 mm film, as mentioned within the technical fiche of the material provide by manufacturer.”
Additional info on the inks can not be provided as the producers are not disclosing more technical information. The ink recipe is a company secret and is treated as non-disclosed IPR, ensuring thus company’s competitive advantage on the market.
- How did you measure the adhesion between the inks and the substrate? The method and obtained results should be explained in detail.
The evaluation of ink adhesion to substrate, as presented within the manuscript, has been done qualitatively based on imagistic evaluation and based on assessing the continuity of the printed electric traces based on conductivity results (also included within the text).
- SEM, FTIR, AFM etc. are required to show the physical and chemical changes of the surface and explain the obtained improvements in printing results.
It is definetely true that all the aforementioned analysis (SEM, FTIR si AFM) are providing important information on the evaluation on conductive inks printing or patterning, at the level of PTFE, but our main goal within the herein study was mainly to demonstrate the efficiency of non-treatment effect on PTFE. The effect of non-thermal plasma treatment at the level of various PTFE surfaces has been done and is already presented within the scientific literature, but, the efficiency of this treatment from functional printing viewpoint has not been discussed in none of them.
Hoping that our responses are in accordance with your question, thank you again for your reply!
Warm regards,
Marius

Round 2
Reviewer 2 Report
Accept.
Moderate English revision is needed.